# Basic Bioelement Contents in Anaerobic Intestinal Sulfate-Reducing Bacteria



**Ivan Kushkevych** [1,*] , **Daryna Abdulina** [2] , **Dani Dordević** [3] , **Monika Rozehnalová** [4] , **Monika Vítězová** [1] , **Martin Černý** [1] , **Pavel Svoboda** [5] and **Simon K.-M. R. Rittmann** [6,*]

1   Department of Experimental Biology, Faculty of Science, Masaryk University, Kamenice 753/5, 62500 Brno, Czech Republic; vitezova@sci.muni.cz (M.V.); cernyarchaea@mail.muni.cz (M.Č.)
2   Department of General and Soil Microbiology, D.K. Zabolotny Institute of Microbiology and Virology of the National Academy of Sciences of Ukraine, Acad. Zabolotnogo Str. 154, 03143 Kyiv, Ukraine; abdulinadarina@gmail.com
3   Department of Plant Origin Food Sciences, Faculty of Veterinary Hygiene and Ecology, University of Veterinary and Pharmaceutical Sciences, 61242 Brno, Czech Republic; dordevicd@vfu.cz
4   Centre of Region Hana for Biotechnological and Agricultural Research, Central Laboratories and Research Support, Faculty of Science, Palacky University Olomouc, 78371 Olomouc, Czech Republic; Jarosova.Monika@email.cz
5   Department of Clinic Subjects, Faculty of Medicine, University of Ostrava, Syllabova 19, 70300 Ostrava, Czech Republic; pavel.svoboda@osu.cz
6   Archaea Physiology & Biotechnology Group, Department of Functional and Evolutionary Ecology, Universität Wien, Althanstraße 14, 1090 Vienna, Austria
*   Correspondence: kushkevych@mail.muni.cz (I.K.); simon.rittmann@univie.ac.at (S.K.-M.R.R.); Tel.: +420-549-495-315 (I.K.); +43-1-4277-76513 (S.K.-M.R.R.)

**Abstract:** The monitoring of trace metals in microbial cells is relevant for diagnosis of inflammatory bowel disease (IBD). Sulfate-reducing bacteria (SRB) represent an important factor in the IBD development. The content of trace metals in bacterial cells may reflect the functioning of the enzyme systems and the environmental impact on the occurrence of SRB. The aim of our research was to compare the content of trace elements in the cells of SRB cultures isolated from fecal samples of patients with IBD and healthy people. The contents of 11 chemical elements in the bacterial cells of SRB were analyzed by the inductively coupled plasma-mass-spectrometry (ICP-MS) method. Significant changes in the content of calcium, zinc, magnesium, potassium, and iron were observed in patients with IBD compared to healthy individuals. Through a principal component analysis (PCA), a total variability of 67.3% in the difference between the samples was explained. The main factors influencing the total variability in the bacterial cells of SRB isolated from patients suffering from IBD were the content of the micro- and trace elements, such as manganese (with power 0.87), magnesium and cobalt (0.86), calcium (0.84), molybdenum (0.81), and iron (0.78). Such changes in the elemental composition of SRB under different conditions of existence in the host may indicate adaptive responses of the microorganisms, including the inclusion of oxidative stress systems, which can lead to changes in SRB metabolism and the manifestation of parameters of IBD in humans. The use of PCA might make it possible in the future to predict the development and ratio of SRB in patients with various diseases.

**Keywords:** trace metals; hydrogen sulfide; toxicity; cell-free extracts; ulcerative colitis

## 1. Introduction

Inflammatory bowel diseases (IBD), which includes Crohn's disease (CD) and ulcerative colitis (UC), occur more often among citizens of European countries, recently [1–5]. The explanation for many diseases is genetic predisposition, causing the impairment of the immune response [4], but the etiology of these diseases remains uncertain. The intestinal microbiome plays a significant role in the intestinal tract [6–14]. Due to their production of the toxic hydrogen sulfide ($H_2S$), sulfate-reducing bacteria (SRB) can significantly influence

the human gut environment [10,15–25]. Therefore, the fact of interaction of microorganisms with a macroorganism—human—is interesting and actual. Our previous research has shown that SRB are involved in the development of IBD [26]. Studies of the intestinal microbiota of patients suffering from UC, especially studies with animal gut inflammation models, demonstrated the connection between SRB and IBD prevalence [15,16,20]. Inflammatory processes under the influence of SRB occur in the human intestine due to the functional role of cellular systems of bacteria, as a result of $H_2S$ and acetate formation [26]. Previous studies have indicated differences between SRB strains isolated from healthy people and individuals with colitis. The differences were observed in the following factors: biomass accumulation, sulfate consumption, lactate oxidation, $H_2S$, and acetate production [27–31]. It is well known that the cytoplasm of bacterial cells, in particular SRB, includes micro- and trace elements. The presence of these elements plays an important role in bacteria enzyme systems (i.e., cytoplasmic, periplasmic, and membrane). Different metals act as cofactors in key enzymes. Several superoxide dismutases depending on the type of metal cofactor of the active center of enzymes: Cu, Zn–SOD, and Mn–SOD, as well as the less common Fe–SOD and Ni–SOD [32–34]. ATP sulfurylase, an enzyme of the process of dissimilatory sulfate reduction, contains Zn [35]. APS (adenosine-phosphosulfate) reductases and dissimilatory sulfite reductases preferably contain $Fe^{2+}$ or $Fe^{3+}$ [36]. Enzymes involved in lactate oxidation, in particular lactate dehydrogenase, function together with cytochrome $c3$, which contains iron ions. Pyruvate:ferredoxin oxidoreductase contains thiamine diphosphate and [4Fe–4S] clusters. Acetate kinase, for example, requires $Mg^{2+}$ for their activity [9].

Elements such as $Na^+$, $K^+$, $Ca^{2+}$, and $Mg^{2+}$ are non-cofactors and play a substantial role in the sodium potassium pump and transport to the cytoplasm, same as in the synthesis of ATP in SRB. Additionally, antioxidant enzymes (superoxide dismutase (SOD) EC 1.15.1.1, catalase EC 1.11.1.6, and peroxidase EC 1.11.1.7) are important for SRB because they are involved in the stress response of bacterial cells. Catalase has sites for binding to metal ions, iron (Fe) ions, heme, and NADP. Thus, a lot of trace metallic elements are involved in SRB metabolism and are essential for bacterial cells.

Despite the essential role in bacterial enzymatic systems of bacteria, some metals (i.e., Pb, Co, Cu) occurring at high concentrations, are often toxic for biological systems, mainly through the following effects [37]: (a) the blockage of enzymes' and transport systems' function in all groups; (b) essential ions can be replaced by the present metals; (c) the active conformation of biomolecules can be modified in the presence of metals. Metal ions bond with proteins, nucleotides, coenzymes, phospholipids, porphyrins, and other metabolites due to their capability to form complexes with hydroxyl, phosphate, carboxyl, amino, and sulfhydryl groups [38]. The toxicity of metals can be overviewed through their adverse effects on cell transport functions; they can have mutagenic effects. Additionally, toxic metals can inhibit DNA replication, cause respiratory depression and cytoplasmic dysfunction, and can influence nitrogen fixation processes. Synthesis of RNA, protein, riboflavin, and vitamin $B_{12}$ can also be affected by the presence of some metal ions [39].

Thus, the content of micro- and trace elements in the bacterial cells could indirectly reflect the functioning of the enzyme systems and bacterial cells in general. Changes in the element's composition inside bacterial cells can be caused by changes in the environment, i.e., in the physiological state of patients, including stress, changes in diet, and the appearance of excess weight. Consequently, the presence of different metals inside the bacterial cell means the changes in the environments for enzymes and physiological changes in bacteria can affect the occurrence and prevalence of some IBDs. However, the trace-element content of the bacterial cell will not only reflect the effect of human body conditions. Therefore, the aim of the research was to compare the content of trace elements in the bacterial cells of SRB cultures isolated from fecal samples of patients suffering from IBD and healthy people.

## 2. Materials and Methods

### 2.1. The Samples

The fecal samples from patients suffering of IBD, i.e., ulcerative colitis (UC) and Crohn's disease CD (10 samples) and healthy people (4 samples) as a control group were collected during a period between 2005–2018. The features of the patients and healthy individuals, which were used in this study are presented in Table 1. The study was conducted in accordance with the Declaration of Helsinki. The protocol was approved by the Ethics Committee of University Hospital Ostrava and Masaryk University (551/2018). The samples from the 10 patients suffering from IBD and the specimens from the healthy people were obtained at the University Hospital of Ostrava (Czech Republic). The subjects of the research were informed about the use of their samples for the research project and they agreed to this use and signed a consent form, which was obtained from each of the individuals.

**Table 1.** Patients and control subject's characteristics. IBD: inflammatory bowel disease.

| Patient | Sex (M/F) | Diagnosis | Age (Years) | Weight (kg) | Year of Diagnosis | State of the Disease * | Isolated SRB Strains |
|---|---|---|---|---|---|---|---|
| | | | | Diseased persons | | | |
| Patient 1 | M | CD | 24 | 73 | 2006 | Long-term remission | *D. vulgaris* M9.1 *Desulfovibrio* sp. M9.2 |
| Patient 2 | F | CD | 21 | 57 | 2011 | Short-term remission | *D. vulgaris* M9.3 SRB M1.1 *D. vulgaris* M1.2 |
| Patient 3 | M | CD | 34 | 104 | 2013 | Short-term remission | SRB M2.1 SRB M2.2 SRB M2.3 |
| Patient 4 | M | CD | 34 | 70 | 2018 | Disease flare | *D. vulgaris* M10.1 SRB M10.2 |
| Patient 5 | M | UC | 44 | 61 | 2014 | Long-term remission | *D. vulgaris* M7.1 *D. vulgaris* M7.2 *D. vulgaris* M7.3 |
| Patient 6 | M | UC | 35 | 92 | 2018 | Short-term remission | *Desulfovibrio* sp. M5.1 *D. vulgaris* M5.2 |
| Patient 7 | F | UC | 20 | 70 | 2003 | Short-term remission | *D. vulgaris* M8 |
| Patient 8 | F | UC | 66 | 78 | 2012 | Short-term remission | *D. vulgaris* M3.1 *D. vulgaris* M3.2 |
| Patient 9 | F | UC | 41 | 53 | 2005 | Disease flare | SRB M4.1 SRB M4.2 *D. vulgaris* M4.3 |
| Patient 10 | F | UC | 39 | 120 | 2013 | Disease flare | *D. vulgaris* M6.1 *Desulfovibrio* sp. M6.2 SRB M6.3 |
| | | | | Healthy persons | | | |
| Control 1 | M | no IBD | 46 | 108 | – | Healthy | SRB Z8.1 *D. vulgaris* Z8.2 |
| Control 2 | M | no IBD | 53 | 72 | – | Healthy | SRB Z9.1 *D. vulgaris* Z9.2 *D. vulgaris* Z9.3 |
| Control 3 | M | no IBD | 42 | 95 | – | Healthy | *D. vulgaris* Z11.1 SRB Z11.2 SRB Z11.3 |
| Control 4 | M | no IBD | 26 | 78 | – | Healthy | *D. vulgaris* Z12.1 SRB Z12.2SRB Z12.3 |

* The induction of remission is dated from for each sample as followed: 5/2019 (M-01), 10/2019 (M-02), 4/2019 (M-03), 10/2019 (M-05), 2/2015 (M-07), 7/2019 (M-08), 11/2017 (M-09).

## 2.2. Bacterial Strains

All the bacterial strains were isolated, purified, and identified in previous research [6]. PCR products were sent to sequence analysis of the 16S rDNA gene. 16S rDNA genes of SRB were deposed in GenBank under the following accession numbers: MT027899, MT093800, MT093826, MT093820, MT093819, MT093823, MT093825, MT093830, MT093829, MT093831. The strains were deposited in the collection of microorganisms of the Laboratory of Anaerobic Microorganisms, which is situated in the Department of Experimental Biology of the Masaryk University (Brno, Czech Republic).

## 2.3. Bacterial Cultivation

The intestinal sulfate-reducing bacteria (SRB) were grown in an SRB medium, as modified by Kováč and Kushkevych (2016) [40]. The experimental approach was based on the cultivation of SRB in Postgate medium [41]. An ascorbic acid solution of 10% ($w/v$) was added into the growth medium to a final concentration of 0.1 g $L^{-1}$. Ascorbic acid served as reducing agent to decrease the redox potential ($E_h$ = −100 mV) [42]. For the detection of colonies, a 10% ($w/v$) of Mohr's salt solution $(NH_4)_2Fe(SO_4) \cdot 6H_2O$ was added to a final amount of 10 mL $L^{-1}$. Iron ions included in Mohr's salt react with $H_2S$ to produce a black precipitate (FeS) upon growth of SRB, which allows visualizing their colonies. After completion, the pH of the growth medium was adjusted to approximately 7.5 with a 1 mol $L^{-1}$ NaOH solution. For the cultivation in Petri dishes, the same composition of the medium was used, with the addition of nutrient agar at a concentration of 20 g $L^{-1}$ before sterilization was performed.

## 2.4. Morphology of Bacterial Cells

1 mL of a *Desulfovibrio* suspension was filtered through a polycarbonate membrane filter (0.2 μm, Millipore, Burlington, Massachusetts, USA) using a vacuum filtration device (Millipore, Burlington, Massachusetts, USA). Samples were fixed by 2% ($v/v$) glutaraldehyde in 0.1 mol $L^{-1}$ sodium cacodylate buffer (pH 7.4). After 1 h at room temperature, overnight fixation and subsequent removal of fixative, the samples were immediately transferred to 50% ($v/v$) ethanol solution and incubated at 4 °C until further processing. Thereafter, a small piece of filter with bacteria on it was cut out and dehydrated by ethanol solution series using an ethanol concentration of 70, 85, 95, and 100% ($v/v$). Each dehydration step lasted approximately 20 min at room temperature. The 100% ($v/v$) ethanol saturated filters were then dried at the critical point with liquid $CO_2$ and put on stub conductive carbon tape. To further increase their conductivity, the samples were sputter-coated by using 2 nm of platinum. Afterwards, the samples were analyzed using a field emission scanning electron microscope TESCAN MIRA3 (TESCAN, Brno, Czech Republic) at 50,000–100,000 times instrumental magnification. The images were obtained at 3 keV accelerating voltage using an In-Beam secondary electron detector.

## 2.5. Measurement of Microelements in Bacterial Cells

The bacterial strains were collected from the end of the exponential growth phase (48 h) and washed three times in a phosphate buffer. The accumulated biomass (1.5 mL) was disintegrated by an ultrasonic device and vortexed prior to the microwave digestion in an MLS 1200 Mega closed vessel digestion unit (Milestone S.r.L., Sorisole, Italy) using a mixture of 2 mL $HNO_3$ and 0.5 mL of $H_2O_2$. A power-controlled digestion program was applied: 2 min–250 W, 2 min–0 W, 5 min–400 W, 2 min–0 W, 2 min–500 W, 2 min–0 W, and 6 min–600 W. The digests were then cooled to laboratory temperature, diluted with ultrapure water in a 10 mL volumetric flask. The blank samples underwent the same procedure as the bacterial samples. To quantify the content of the trace elements, two series of calibration solutions were prepared by diluting single element certified reference materials in ultrapure water. The calibration range was 0.1–1000 μg $L^{-1}$ for Co, Cu, Fe, Mo, Mn, Ni, and Zn and 1–10 000 μg $L^{-1}$ for Ca, K, and Mg, respectively. All used plastic materials and

volumetric glassware were placed into a 10% ($v/v$) nitric acid bath for 24 h and were then rinsed three times using ultrapure water and dried.

The quantitative analysis of chemical elements was carried out by the inductively coupled plasma-mass-spectrometry (ICP-MS) using a 7700× instrument (Agilent Technologies, Tokyo, Japan) equipped with an octopole reaction/collision cell, a quadrupole mass analyzer, and an ASX-520 autosampler. The optimized ICP-MS instrumental conditions for He mode were as follows: RF power of 1550 W, plasma gas flow rate of 15.00 L min$^{-1}$, auxiliary gas flow rate of 0.9 L min$^{-1}$, nebulizer gas flow rate of 1.05 L min$^{-1}$, and dwell time 100 ms for all measured isotopes ($^{24}$Mg, $^{39}$K, $^{43}$Ca, $^{55}$Mn, $^{56}$Fe, $^{59}$Co, $^{6O}$Ni, $^{63}$Cu, $^{66}$Zn, and $^{95}$Mo) including internal standards ($^{45}$Sc, $^{72}$Ge, $^{89}$Y, and $^{115}$In). Sodium (Na) content was measured by flame spectrometer BWB-EP (BWB Technologies, Newbury, the UK) and the calibration range was 5–1000 mg L$^{-1}$. Each bacterial sample was measured in six replicates, and the results are presented as an arithmetic mean ± SD, where SD denotes the standard deviation.

Measurement accuracy was verified by the repeated analysis of a quality control sample at the concentration of 100 µg L$^{-1}$ for macro elements; the accuracy for micro elements was tested by the analysis of external certified reference materials (CRM) of water TMDA-64.3 and TM-25.6. Before the application of the method for real bacterial sample analysis, the ICP-MS method was validated by using spike samples at the concentration level of 10 µg g$^{-1}$ for Mn, Co, Ni, Cu, and Mo; 100 µg g$^{-1}$ for Mg, Fe, and Zn; and 1 mg g$^{-1}$ for K and Ca.

*2.6. Statistical Analysis*

The analysis of the obtained results was performed by using Principal Component Analysis (PCA) [43,44]. The analysis determined the main components, their eigenvalue, matrixes of loadings, and scores. The analysis was performed by nonlinear iterative projection using alternating least squares (NIPALS—non-linear iterative projections by alternating least-squares); the number of iterations was 9–11 cycles [45]. Statistical analysis was performed using MS Office (2010), Origin 8.0 (https://www.originlab.com/) and Statistica 13 (http://www.statsoft.com/). Statistical parameters (mean: M, standard error: m, M ± SE) were obtained. The reliability of the mean values for metal concentrations was performed for small samples (N = 10–14) was performed using Student's *t* test at $p \leq 0.05$ [46].

**3. Results**

In our previous research, SRB strains were isolated and identified to belong to the *Desulfovibrionaceae* [47]. Figure 1 shows SRB strains isolated from IBD patients and healthy individuals (control). However, no significant difference in cell morphology was observed. Therefore, the content of 11 micro- and trace elements in the bacterial cells of SRB isolated from the studied patients were investigated. Figure 2 and Table 2 shows the contents of 11 elements. According to the data obtained from the content of chemical elements in the bacterial cells of SRB, isolated from patients with IBD and healthy, differences are visible for all the metals. In bacteria from the control group of SRB was noted the presence of elements such as sodium 3.58–4.61 µg g$^{-1}$, potassium 15.1–22.8 µg g$^{-1}$, calcium 4.6–22.1 µg g$^{-1}$, magnesium 0.61–1.21 µg g$^{-1}$, zinc 0–0.55 µg g$^{-1}$, iron 1.16–18.15 µg g$^{-1}$, and trace amounts of copper, manganese, cobalt, nickel, and molybdenum. Significant changes were observed in SRB cells from Crohn's disease patients compared to the control group in calcium content (range of concentrations calcium: 3.0–55.6 µg g$^{-1}$; zinc: 0–0.88 µg g$^{-1}$; magnesium: 0.77–1.69 µg g$^{-1}$; potassium: 16.3–27.9 µg g$^{-1}$). The content of the bioelements inside the SRB were increased by almost 2.5, 1.6, 1.39, and 1.2 times, respectively.

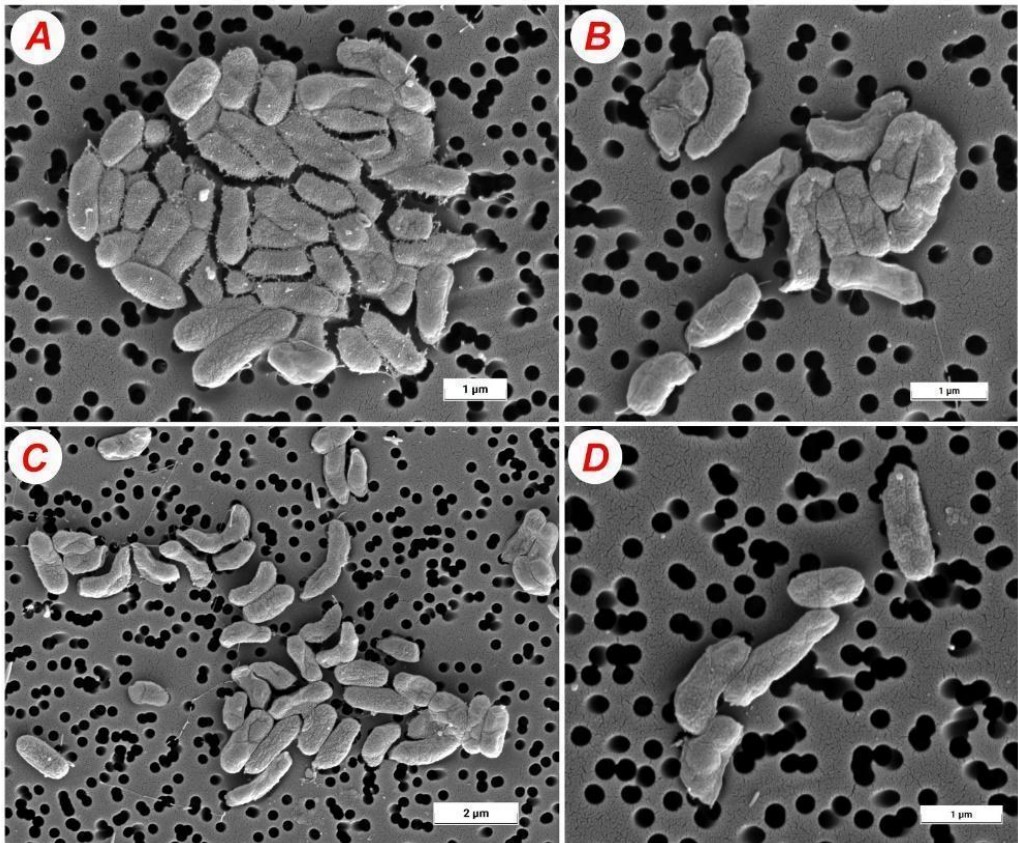

**Figure 1.** The morphology of *Desulfovibrio* cells from patients with IBD. Note. Strains from patient with Crohn's disease: *D. vulgaris* M1.2 (**A**), ulcerative colitis: *D. vulgaris* M3.1 (**B**) and control (healthy): *D. vulgaris* Z8.2 (**C**) and *D. vulgaris* Z11.1. (**D**).

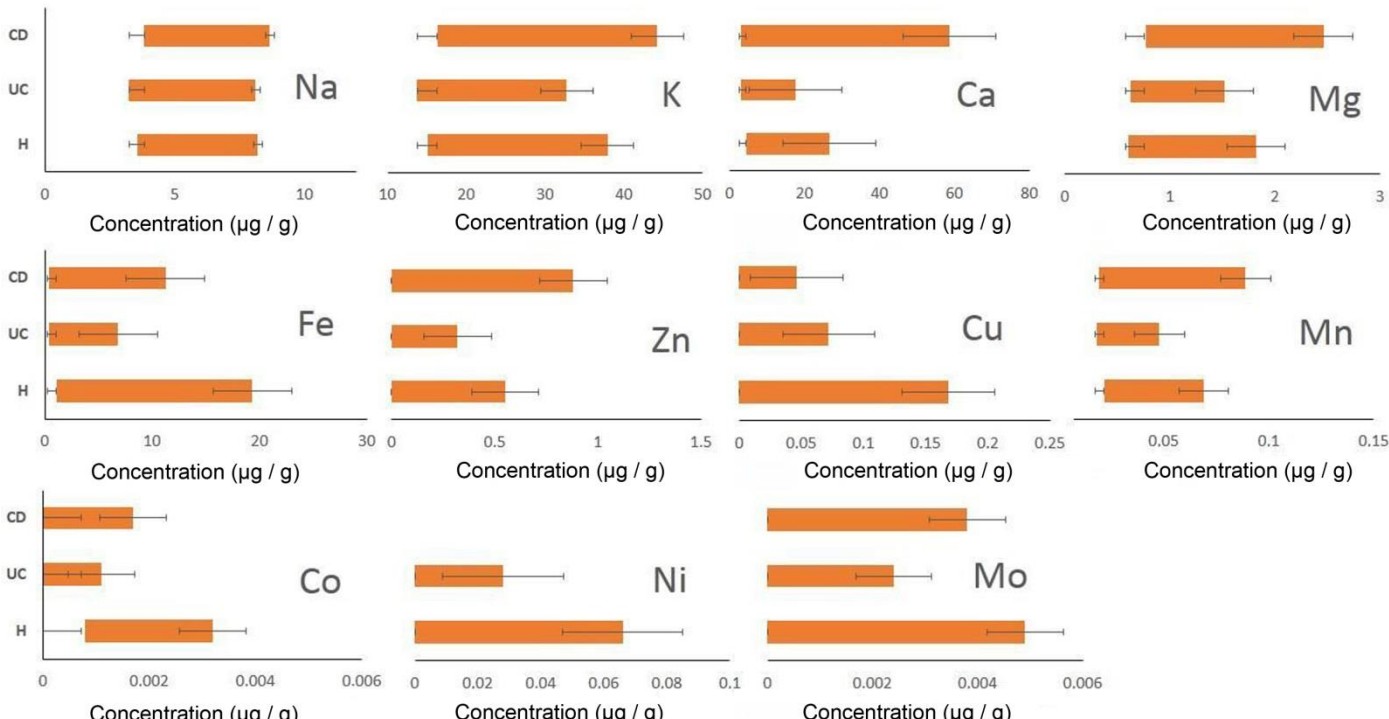

**Figure 2.** The content of elements in the sulfate-reducing bacteria SRB isolated from patients suffering from IBD. Notes: Crohn's disease (CD), ulcerative colitis (UC), healthy (H).

**Table 2.** Significance of the mean values of the metal concentrations in the cytoplasm of SRB from patients with IBD.

| Groups | Parameters | $Na^+$ | $K^+$ | $Ca^{2+}$ | $Mg^{2+}$ | $Zn^{2+}$ | $Fe^{2+}$ | $Cu^+$ | $Mn^{2+}$ | $Co^{2+}$ | $Ni^{2+}$ | $Mo^{2+}$ |
|---|---|---|---|---|---|---|---|---|---|---|---|---|
| CD | X | **4.46** | **21.84** | **15.66** | **1.07** | 0.17 | 3.96 | **0.02** | **0.033** | 0.0005 | 0 | 0.0011 |
| | Δx (t lim 2.26, n = 9) | 0.22 | 3.17 | 13.31 | 0.24 | 0.22 | **2.62** | 0.01 | 0.01 | 0.0004 | 0 | 0.0011 |
| UC | X | **4.22** | **17.30** | **7.36** | **0.77** | **0.08** | **1.47** | **0.023** | **0.027** | 0.0005 | 0.0052 | **0.0001** |
| | Δx (t lim 2.16, n = 14) | 0.28 | 0.89 | **2.29** | 0.06 | 0.06 | 1.02 | 0.01 | 0.002 | 0.0002 | 0.0061 | 0.0003 |
| H | X | 4.16 | **18.37** | **11.41** | **0.94** | **0.25** | 8.41 | **0.04** | **0.03** | 0.001 | 0.01 | 0.0022 |
| | Δx (t lim 2.23, n = 11) | 0.24 | 1.38 | **3.67** | 0.13 | 0.11 | **3.69** | 0.039 | 0.004 | 0.0003 | 0.0144 | 0.0010 |

Note: Crohn's disease (CD), ulcerative colitis (UC), healthy (H), X is average mean. Inaccurate data according to Student's test (*t*) at $p \leq 0.05$ are in bold.

The content of iron was 0.4–10.86 µg g$^{-1}$ and decreased 1.67 times. For SRB strains isolated from patients suffering from UC, the content of iron was observed in the concentration 0.42–6.42 µg g$^{-1}$, which is 2.82 times less than the healthy group. The contents of zinc, calcium, magnesium, and potassium were decreased 1.71, 1.51, 1.25, and 1.2 times, respectively. It was noted that the sodium contents in all three groups of SRB strains fluctuated slightly. Significant variations in calcium content may be due to changes in the functioning of the cytoplasmic membrane, where this element is involved. Changes in the content of copper, manganese, cobalt, nickel, and molybdenum were also noted, but insignificant. However, this does not exclude the role of these elements in the functioning of SRB cells under different conditions of IBD.

To determine the influence of the elements' composition on the distribution of SRB strains, cluster analysis by the PCA method was performed. This method is widely used for the analysis of multidimensional systems in biology, physics, medicine, etc. [44,45,48,49].

PCA is based on three components: scores, loadings, and explained variations. For the analysis, we used data on the content of 11 chemical elements in the cells of 35 strains of SRB isolated from patients with IBD and the control healthy persons. Characteristics of the three obtained principal components are shown in Table 3.

**Table 3.** PCA results and explained variations.

| Importance | Factors | Power | Principal Components | | |
|---|---|---|---|---|---|
| | | | PC1 | PC2 | PC3 |
| 1 | Mn | 0.87 | 0.874 | 0.333 | 0.068 |
| 2 | Mg | 0.86 | 0.836 | 0.388 | −0.138 |
| 3 | Co | 0.86 | 0.729 | −0.456 | 0.3411 |
| 4 | Ca | 0.84 | 0.718 | 0.337 | −0.468 |
| 5 | Mo | 0.81 | 0.806 | −0.323 | 0.233 |
| 6 | Fe | 0.78 | 0.765 | −0.317 | 0.297 |
| 7 | Na | 0.77 | 0.066 | 0.460 | 0.753 |
| 8 | Cu | 0.76 | 0.565 | −0.598 | −0.279 |
| 9 | K | 0.72 | 0.475 | 0.701 | −0.074 |
| 10 | Zn | 0.68 | 0.796 | 0.200 | −0.121 |
| 11 | Ni | 0.59 | 0.296 | −0.699 | −0.110 |
| | Eigenvalues | | 5.02 (S) | 2.38 (S) | 1.18 |
| | Total variance (%) | | 45.65 | 21.65 | 10.74 |

Note: values highlighted in red mean that the loadings of this factor are insignificant, the sign "−" determines the position of the point in the coordinate system, S is significant.

The first principal component, PC1, is significant with an eigenvalue of 5.02 and explains up to 45.65% of the variations. Therefore, the main factors influencing the variations in the SRB isolated from patients with IBD are the content of the micro- and trace elements such as manganese (with power 0.87), magnesium and cobalt (0.86), calcium (0.84), molybdenum (0.81), and iron (0.78). That is, within the main component, the greatest

contribution to the distribution of samples of SRB strains is manganese. The second major component of PC2 also has a significant eigenvalue (2.38) and explains up to 21.65% of the variations. The second component is loaded with elements such as potassium and magnesium, while negative correlations were observed for copper, nickel, and cobalt metals, although they exerted a significant load. The third component explained up to 10.74% of the variations, and sodium (power 0.77) put a significant load on it. On the plot, each variable is represented by a point in coordinates (PC1, PC2). Analyzing it similarly to the score plot, we can observe which variables are closely related and which are independent [44]. However, the most informative joint study of paired plots of scores and loadings (Figure 3).

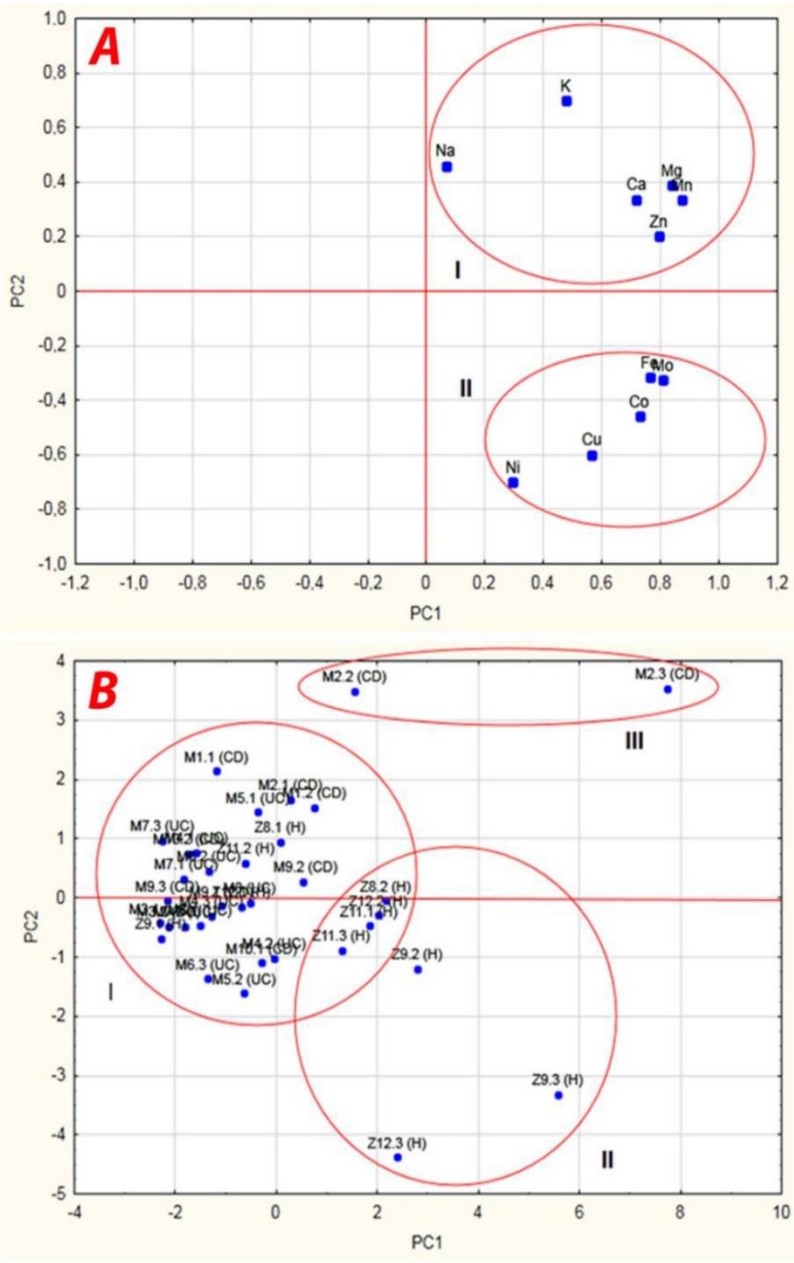

**Figure 3.** Loading (**A**) and score (**B**) plots of the PCA for samples from patients with IBD. Note: designation of SRB strains as in Table 1; UC indicates ulcerative colitis, CD indicates Crohn's disease, and H indicates a healthy person.

Figure 3A shows the grouping of chemical elements into two clusters, the first cluster includes elements such as manganese, magnesium, zinc, and cobalt. Sodium and calcium

in this cluster are slightly separated. This is probably due to their low importance in the factor distribution (powers were 0.77 and 0.72, respectively). The second cluster contains elements such as iron, molybdenum, cobalt, copper and nickel, all of which belong to the transition group of chemical elements and almost all of them are involved in the enzymatic systems of SRB cells, as described above.

Figure 3B shows the distribution of SRB samples by elemental composition on the principal components PC1 and PC2, which together explained the total variations of the difference between the samples by 67.3%. In the score plot, all points located along the axes of the principal components form three groups. The largest group, group I, included SRB isolated from patients with UC and CD. The second group, group II, has a transition zone with group 1 and includes three strains of SRB Z.9.2, Z9.3, and Z12.3, which were isolated from healthy people. It should also be noted that between the two groups, there is a transitional cluster of SRB, which are isolated from both healthy people (Z8.2., Z11.1, Z11.3, Z12.2) and patients with IBD. Group III includes two strains of SRB, namely M.2.2 and M2.3. These bacterial strains were isolated from patient M-02 with CD in a state of short-term remission. The peculiarity of this patient is overweight (up to 104 kg); we can once again emphasize the probability of the impact of this indicator on the development of IBD.

It should be noted that the characteristics of patients to the analysis matrix were not considered. The strains of SRB are located in the middle of the two groups isolated from healthy individuals; however, according to the elemental composition, they are similar to the strains isolated from patients. SRB strains Z8.2., Z11.1, and Z11.3, isolated from individuals with an increased weight of 95–108 kg, it is likely that these people may be at risk, because in the SRB cells, there have been changes that bring them closer to the same composition as in SRB strains from patients with IBD.

The position of cluster II is close to the cluster of patients with IBD, which indicates that even a slight adverse effect on the patient's body can cause irreversible changes in human health. The use of factor analysis makes it possible to further predict the development and the ratio of SRB in patients with various diseases.

## 4. Discussion

The daily human diet may contain a high amount of sulfate if certain food commodities are consumed, such as those containing sulfur oxides, sulfate polysaccharides (mucin), chondroitin sulfate, and carrageenan. The calculated sulfate intake in the western diet may be as high as 16.6 mmol sulfate per day [50]. Hence, there is a higher probability that the feces of healthy individuals contain SRB (*Desulfovibrio*: up to 92%) [51–53]. We would like to emphasize again that high $H_2S$ concentrations are also toxic for the host. Growth of *Desulfovibrio* sp. commences at $H_2S$ concentrations of >6 mmol $L^{-1}$; however, the microbial metabolic activity is not inhibited by a 100% concentration. The lag phase and the generation time of *Desufovibrio* sp. at $H_2S$ concentrations of 5 mmol $L^{-1}$ lasted two times longer and was found to be eight times higher, respectively, compared to optimal growth conditions of these organisms [17]. Clostridia are also known to produce $H_2S$; however, only in smaller quantities [54]. Studies using mice showed a 1.14 times higher $H_2S$ production among mice with UC, in comparison with a mouse control group [17,55,56]. Concerning the development of the intestinal diseases, sulfate consumption, lactate consumption, $H_2S$ production, and accumulation of acetate play an important role [57]. *Desulfovibrio* sp. are related to the development and occurrence of IBD, since they can be found in the intestine and feces of affected people and animals. SRB use sulfate as a terminal electron acceptor [57–61]. The sulfate present in different food commodities, such as some bread, dried fruits, soy flour, brassicas, and sausages, and in some beers, wines, and ciders [50], could be an important factor with regard to the development of these diseases.

Zinc, nickel, cobalt, strontium, copper, lead, cadmium, and uranium are mainly transported into the cell, but some of the ions bind to its surface. The active supply of these

metals, as a rule, is carried out by the $Mg^{2+}$ transport system and—much less often—$Mn^{2+}$ and $Ca^{2+}$ due to the stereochemical analogy [62]. As a result of the adaptation of microbial cells to heavy metals, the accumulation of the latter increases by 1.2–2.2 times. It is believed that bacteria that inhabit ecological niches with a high content of toxicant absorb it in higher concentrations than those isolated from areas with low concentrations of metal. There are three groups of values of the concentration of heavy metals acting on microbial cells. Low concentrations stimulate the development of bacteria due to the Arndt-Schultz effect. Violation of the barrier function of the membrane promotes the entry of nutrients into the cell and increases metabolism. Lead ions at a concentration of up to 0.5 mmol $L^{-1}$ can increase the specific growth rate of soil bacteria [63]. Excess copper leads to inhibition of cell metabolism. Additionally, adding sublethal concentrations of heavy metals, such as cadmium, into the nutrient medium during long-term storage of bacteria helps maintain a certain level of resistance to this toxicant [64]. The degree of influence of heavy metals on microorganisms is, e.g., determined by physicochemical environmental factors, such as pH, characteristics of toxicant salts, and the microbial metabolism [65,66].

The SRB *Desulfotomaculum reducens* sp. nov. MI-1 may utilize various sulfur compounds as electron acceptors, but also Cr (VI), Mn (IV), Fe (III), and U (VI) [67]. A characteristic feature of strain MI-1 is that it uses metals as an electron acceptor in the absence of sulfate. With butyrate as a carbon source, strain MI-1 is able to grow using Mn (IV), Fe (III), U (VI), or Cr (VI) in the absence of sulfate. Other electron donors, such as lactate or valerate, are also known to support its growth. It was found that SRB can reduce metals enzymatically or through sulfate reduction with the formation of $H_2S$, and therefore, low concentrations of heavy metals, such as Cr (VI), Hg (II), and Zn (II), enhanced the growth and sulfate-reducing activity of SRB [68,69]. Tebo and Obraztsova first showed that SRB grow by oxidizing organic matter and reducing Cr (VI) to Cr (III), Mn (IV) to Mn (II), Fe (III) to Fe (II), and U (VI) to U (IV), and demonstrated that the reduction of Cr (VI) supports growth [67]. Our study of the mechanisms involved in the reduction of metals might be relevant for understanding how trace metals are reduced, whether there is a hierarchical advantage of the electron acceptor, and how cells become tolerant to toxic metals. Studies that deal with SRB presence in the intestines and correlation about the prevalence of IBD and $H_2S$ are very important since $H_2S$-releasing compounds can be regarded as therapeutic agents [70]. $H_2S$ was also confirmed to be an important cardiovascular as well as nervous system signaling factor. Through the conversion to thiosulfate, the cecal mucosa protects itself from toxic effects of $H_2S$ [71,72].

The results obtained by us using PCA allowed us to determine the load on the distribution of SRB strains by elemental composition. The sum of the total variation was 67.3%. The use of PCA has been shown to be effective in a number of clinical and environmental studies [43,44,72]. As a result of applying the principal components method for processing an array of medical data on the content of a number of chemical elements in human biological substrates for people living and working in the study area, it was shown that certain patient groups are visually identified using the billing graphs, which have different indicators for health reasons. Moreover, it was proven that by varying the components, the similarities or differences between patients can be identified, which depend on the concentration in the blood or hair of certain chemical elements. The application of PCA allows us to conclude the microelement state of the human body ("microelementoses"—a concept that understands "states of deficiency, excess or imbalance of macro-micronutrients that naturally affect human health"—introduced in an earlier study [44]).

Environmental studies also applied PCA. In the research by Abdulina (2016) [43], to analyze the relationships between the number of microorganisms of various ecological and trophic groups of the sulfidogenic community and some environmental parameters of their environment, a PCA was carried out. For the analyzed system with 12 variables, 5 main variables were identified by the values of the explained variance components, explaining up to 76% of the variations. The first main component (PC1), which includes such variables as the number of iron-reducing and ammonifying bacteria, explained up

to 21.5% of the variations. The second main component (PC2) explained 18.5% of the variations and contains variables such as pH and the amount of nitrogen-fixing and denitrifying bacteria. The third component (PC3) explains up to 14.1% of the variations; the load on it is carried out by the temperature factor and the content of soluble sulfates. The number of SRB accounts for up to 11.4% of the variations (component PC4). The exception was the variable "number of SRB," for which the load on PC4 was 0.72, and therefore, it might be possible that these bacteria exhibit a key role in the sulfidogenic community [43].

In the work by Eiler (2003) [48], five diluted soda lakes in eastern Austria and factors of abundance of heterotrophic bacteria, their production, and their controlling factors were investigated. In these ecosystems, the environmental factors, which are responsible for the control of the microbial community in the shallow soda pools, were investigated during an annual cycle. Through PCA, two factors were extracted, which explained up to 62.5% of the total variation of the ecosystems. The first factor might be interpreted as the turbidity factor. The second factor was referred to as a concentration factor. From this analysis, it was deduced that the bacterial and cyanobacterial abundance in these ecosystems were mainly controlled by resuspension through wind-induced sediment, and that the turbidity was stabilized by the high pH and salinity, and rather less so by dissolved organic carbon and the evaporative concentration increase of salinity. Furthermore, bacterial production clustered together with temperature [48].

According to the results of the PCA, it was determined that the main load on the distribution of SRB strains isolated from patients with IBD were transition metals such as manganese, magnesium, cobalt, calcium, molybdenum, and iron. From the literature, it is known that some SRB can reduce Fe (III), U (IV), and Cr (VI) [71,73]. The accumulation of heavy metals in bacterial cells can be carried out by ion transport systems ($Ca^{2+}$, $Na^+$, $K^+$, etc.), which are necessary for the normal functioning of the cell [31,74,75]. Our study shows that an increase in calcium, which was observed in SRB cells isolated from IBD patients, may indicate changes in the functioning of SRB cell transport systems, which is known to be located on the membrane in prokaryotic cells [76]. The cytoplasmic membrane, in turn, is involved in the antioxidant defense system of the cell [77]. Additionally, the PCA noted that overweight people, despite belonging to the group of healthy individuals, had SRB with a cell content similar to patients with IBD. Though, the samples from the both groups (healthy and not healthy individuals) were obtained from feces, and these results can be described as findings in the normalized laboratory environment. These findings in the decrease in soluble ferric ions in SRB isolated from IBD patients for both CD and UC may indicate the formation of $H_2S$ and insoluble sulfides due to the intensification of dissimilatory sulfate reduction processes.

Therefore, the question of the interaction of macro- and microorganisms remains interesting. For example, a question whether the macroorganism with IBD is a prerequisite for natural selection of SRB strains is still open. The trace-element contents of the bacteria also could reflect the expression of the genes, in normalized conditions and not in the human body. It is also interesting to study the changes at the genetic level in SRB, which are isolated from patients suffering from IBD and healthy people. According to the results of PCA, changes in the concentration of some trace elements could, therefore, be interpreted as a marker of specific genomes adapted to IBD. Whether they can be used as a marker of a specific proteomes adapted to IBD is still unknown.

## 5. Conclusions

SRB were shown to be present in the samples of healthy and individuals suffering from IBD. However, statistically significant ($p < 0.05$) differences could be observed among all measured parameters. The main trace elements that distinguished the samples, according to the detected chemical element, were calcium, manganese, zinc, potassium, and iron. The principal components were determined, and they explained the total variations of the difference between the samples up to 67.3%. The main factors influencing

the total variations in the SRB isolated from patients with IBD are content of the micro- and trace elements such as manganese (with power 0.87), magnesium and cobalt (0.86), calcium (0.84), molybdenum (0.81), and iron (0.78). Such changes in the elemental composition, especially for transition metals (iron, cobalt, and molybdenum) of SRB under different conditions of existence in the macroorganism, may indicate adaptive responses, including the inclusion of oxidative stress systems, which may lead to changes in SRB metabolism, including intensification sulfate reduction due to stress, $H_2S$ synthesis, and the manifestation of the parameters of disease in humans. The physiological differences among the examined SRB strains of healthy individuals and individuals suffering from IBD emphasized the importance to further understand the associated microbiological processes that are involved in IBD. The use of factor analysis makes it possible to further predict the development and ratio of SRB in patients with various diseases.

**Author Contributions:** Conceptualization, I.K., D.A. and M.V.; methodology, I.K., M.Č., P.S. and M.R.; validation, I.K., M.V. and D.D.; formal analysis, M.V. and S.K.-M.R.R.; investigation, I.K., M.Č.,P.S. and M.R.; resources, I.K.; data curation, M.V. and I.K.; writing original draft preparation, I.K., D.A., D.D., M.V. and S.K.-M.R.R.; writing review and editing, I.K., D.A., D.D. and S.K.-M.R.R.; visualization, I.K. and D.A.; supervision, M.V.; project administration, I.K.; funding acquisition, S.K.-M.R.R., M.V., D.D. and I.K. All authors have read and agreed to the published version of the manuscript.

**Funding:** This research was supported by the Grant Agency of Masaryk University (MUNI/A/1425/2020); open access funding by the University of Vienna.

**Institutional Review Board Statement:** Not applicable.

**Informed Consent Statement:** Not applicable.

**Data Availability Statement:** Not applicable.

**Acknowledgments:** The authors express their gratitude to Iva Buriánková for the transportation of the samples as well as to Jakub Javůrek and Markéta Machálková from Demo Lab & Applications TESCAN ORSAY HOLDING, a.s. Libusinatrida 21623 00 Brno, Czech Republic/Europe, www.tescan.com, for the photographs.

**Conflicts of Interest:** The authors declare no conflict of interest.

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
