# Peer review of "Basic Bioelement Contents in Anaerobic Intestinal Sulfate-Reducing Bacteria"

_applsci, doi:10.3390/app11031152_

Round 1

Reviewer 1 Report

Very informative research. Just the following minor details to amend in the manuscript:

In some of the sentences throughout the manuscript, words are positioned incorrectly in sentences e.g, Line 43-45

Line 77: Start the sentence with "The class of 5' - Adenylylsulfate(APS) Reductases .....". The abbreviation "APS" should not be used the first time it is mentioned in the manuscript. 

Line 324: if you are talking about the concentration of Plumbum please amend the unit of measurement used.

Regarding Table 1.

  1. A definition for a "long -term remission" and "short-term remission" should be provided.
  2. Regarding treatments: did all patients either in remission or experiencing a flare-up receive the same treatment during the "flare-up" episode? There should be a column showing the treatment received and the duration of the treatment during the last flare-up for those in remission and the present treatment for those having a flare-up.
  3. Are those patients who are in remission receiving any probiotics? This could also be put into the treatment column.

Complete the following references details:

  • Give full page range in lines 443, 444, 449, 481, 483, 495, 506, 537, 539, 543, 562;
  • Line 507, the list of authors is incomplete 

Author Response

Dear Reviewer,

We have carefully revised our manuscript and would like to say that we are thankful for your time and your important and critical comments, which helped to improve our manuscript.

We have corrected our manuscript according to your important comments and recommendations.

We have also written our responses to your comments below:

Very informative research. Just the following minor details to amend in the manuscript:

In some of the sentences throughout the manuscript, words are positioned incorrectly in sentences e.g, Line 43-45

It was revised.

Line 77: Start the sentence with "The class of 5' - Adenylylsulfate(APS) Reductases .....". The abbreviation "APS" should not be used the first time it is mentioned in the manuscript.

It was corrected.

Line 324: if you are talking about the concentration of Plumbum please amend the unit of measurement used.

It was revised.

Regarding Table 1.

A definition for a "long -term remission" and "short-term remission" should be provided.

Regarding treatments: did all patients either in remission or experiencing a flare-up receive the same treatment during the "flare-up" episode? There should be a column showing the treatment received and the duration of the treatment during the last flare-up for those in remission and the present treatment for those having a flare-up.

Are those patients who are in remission receiving any probiotics? This could also be put into the treatment column.

The main division of patients was the criterion of presence, absence, or state of recovery. The treatment protocol for the study data was not provided to us. Thus, in the future we plan to study the dynamics of IBD, because it is interesting that it is primary in the occurrence of diseases, the impact of the macroorganism on the microbiota, or vice versa. This information at this stage of research we do not have, unfortunately. In the next stage of the research additional information about patients we will certainly have.

Complete the following references details:

All numbers of pages have been added.

443 ? Give full page range in lines 443, 444, 449, 481, 483, 495, 506, 537, 539, 543, 562;

Line 507, the list of authors is incomplete

All our manuscript has been corrected according to your comments and we kindly ask to you to accept our manuscript for publication.

Best Regards,

Authors

Reviewer 2 Report

General Comment:

The manuscript applsci-1069195-peer-review-v1, entitled “Basic Bioelements Content in Anaerobic Intestinal Sulfate-Reducing Bacteria” by Kushkevych et al. describes the isolation, speciation and analysis of trace elements content of Sulfate Reducing Bacteria, present in fecal samples from patients with Inflammatory Bowel Disease compared to healthy people. After the growth of the respective isolated bacteria, 11 trace elements are quantified by ICP-MS in the bacteria. Based on Principal Component Analysis, the observed changes in the content of these 11 bioelements are proposed, by the authors, as reporters of changes in the metabolism adapted by the SRB in the case of IBD, to specific IBD affected human body environment. The presented work is perfectly executed and the analyses well performed, but I have some concerns about the interpretation the authors made and the ultimate goal of such an analysis. I furthermore consider that the introduction needs to be restructured, with the aim to avoid repetitions, organise the diverse argument and clearly exposing the aim of the study. I suggest therefore the major amendment of the manuscript as developed in more details in specific comments. 

Specific comments:

-L 40-66: I suggest to restructure the whole introduction, avoiding the repetitions, bringing together the distinct facts/data, shortening the text. The idea developed in L52-54 are already exposed in lines 45-47, the idea developed in L54-57 is already present in lines 49-50, the lines 57-59 are also repetitions, lines 59-61 take up ideas already more or less exposed before, the lines 64-66 refer to H2S production already introduced line 48.

-L64-66: There is, as far as I understand, some confusion in the whole manuscript, between the notions of “correlation”, “origins” and “consequences”. In lines 64-66 what does “included” means? Are “H2S production and biomass accumulation of intestinal SRB” only correlated to IBD or established as a cause?

-L67-68: the content of micro- and trace element is not particular 1) to SRB and 2)to cytoplasm.

-L70-94: I suggest to restructure the whole paragraph, by 1) presenting trace elements that are cofactors (distinguishing loosely bound or prosthetic groups, metallic (Fe) or not (Mg+)); 2) presenting trace elements that are not cofactors Na+, K+ and Ca+. The latter act in the cell based on their concentration balance in the cell. All is mixed-up in the present text; 3) referring to a more extended literature. For example, why restricting the list of useful enzymes to superoxide dismutase, catalase, peroxidase ?; 4) articulating the part developed lines 83-93 in a different way since metal as cofactors, are indispensable for a lot of enzymes (as developed just before). I furthermore do not really understand the sentence as it is currently grammatically structured and I consider the use of “On the contrary” line 83, after one paragraph dedicated to metals as cofactors, as inappropriate. 5) presenting the role of trace elements in a more general way. For example, why only citing nitrogen fixation as a process influenced by metals (line 92)?

-L95-102: Here is my main concern. I agree with the authors on the fact that “the content of trace element…may reflect the functioning of the enzyme systems.” (line 95-96) And that, consequently, “the change [in trace elements] can be caused by changes in the environment” (line 97). These changes could in fact reflect “the changes in the environments for enzymes” and thus “physiological changes in bacteria” in some IBD (lines 99-100). But the experimental approach chosen by the authors is to analyze the bacterial cell content after cultivation out of the human body, in a normalized growth medium. The trace-elements content of the bacterial cell will therefore not reflect the effect of human body conditions. The work presented could therefore not be used to establish the metabolic adaptation of SRB to IBD, but could rather be interpreted as the result of a natural selection of SRB strain in the case of IBD. The trace-element contents of the bacteria, as performed in this work, reflect the expression of the genes, in normalized conditions and not in the human body. This content therefore could be interpreted as a marker of specific genomes adapted to IBD (i.e. strain/metabolic selection), but not as a marker of a specific proteomes adapted to IBD (i.e. metabolic adaptation).  

-L209-213: First of all I do not understand the sentence, probably due to the absence of one bracket. But 1) why such variations in bacteria content from similarly infected patients?; 2) why two standard deviations (for minima and maxima). What does it mean? The Usual presentation of such data is an average mean together with a standard deviation. The “increase” or “decrease” “factors” seem to be calculated based on average mean, I therefore suggest to redraw the Fig 2 showing these average mean; 3) I would suggest to present the analysis of data in a more systematic way: for Crohn's disease patients, present the unaffected, the increased and the decreased values, and the same for UC disease patients; 4) I am not convinced by the necessity of the Table 2 in addition of Figure 2; 5) I don't understand why the author consider the Mo content not significantly affected in UC patients. I furthermore suspect an error in the value present in Table 2 for Mo.

-L295-390: I do not really understand the link of these discussion parts with the presented data.

-L391-404: As already exposed I am afraid the authors over-interpreted the results. The presented data only suggest a correlation between “trace element accumulation of bacterial strain” and “microbiota of IBD patients”. This correlation could suggest, and thus be interpreted as, a specific metabolism of bacterial strain present in microbiota of IBD towards trace elements. Rather than reflecting a metabolic adaptation of the isolated strain, this could be interpreted –if the correlation is real- as a metabolic selection, since the analyses of metal content are performed after growth of isolated bacteria in normalized medium rather than directly isolated from feces.

-L405-408: Please could the authors develop more explicitly the interpretation of observed decrease in ferric iron and formation of H2S and insoluble sulfides?

Minor comments:

-L 43-45: please correct the sentence for grammar. I suggest "A significant role in the intestinal tract is played..." or "the intestinal microbiota that ...the macroorganism plays a significant role..."

-L46: please correct the sentence for grammar. I suggest "represent one of the important" or “an important…”

-L52-54: I do not understand the contradiction of the idea exposed in this sentence, implied by “though”, with the preceding one.

-L54: please correct the sentence for grammar. I suggest “SRB are” and since SRB is already in plural, please use “SRB” rather than “SRBs”

-L55, 60, 61: delete unwanted spaces

-L54-57: please correct the sentence for grammar, a verb is missing.

-L70: I do not understand the use of “Certainly” since the authors developed in the following clear data.

-L75-76: Re-write “Cu-SOD” for homogeneity, and please introduce the SOD abbreviation.

-L74 and L80: heme contents iron, thus why distinguishing “iron ion” and “heme” in line 74? if you speak about metal, please, use iron instead of heme in line 80.

-L81-82: what do you mean by “transition elements”?

-L114-122: The distinction between “Bacterial cultures” and “Bacterial cultivation” does not make sense to me.

-L128: “10 mL L-1” is not a concentration unit.

-L324: I do not understand what does it mean, should we read “0.5 mM”?

Author Response

Dear Reviewer,

We have carefully revised our manuscript and would like to say that we are thankful for your time and your important and critical comments, which helped to improve our manuscript.

We have corrected our manuscript according to your important comments and recommendations.

We have also written our responses to your comments below:

General Comment:

The manuscript applsci-1069195-peer-review-v1, entitled “Basic Bioelements Content in Anaerobic Intestinal Sulfate-Reducing Bacteria” by Kushkevych et al. describes the isolation, speciation and analysis of trace elements content of Sulfate Reducing Bacteria, present in fecal samples from patients with Inflammatory Bowel Disease compared to healthy people. After the growth of the respective isolated bacteria, 11 trace elements are quantified by ICP-MS in the bacteria. Based on Principal Component Analysis, the observed changes in the content of these 11 bioelements are proposed, by the authors, as reporters of changes in the metabolism adapted by the SRB in the case of IBD, to specific IBD affected human body environment. The presented work is perfectly executed and the analyses well performed, but I have some concerns about the interpretation the authors made and the ultimate goal of such an analysis. I furthermore consider that the introduction needs to be restructured, with the aim to avoid repetitions, organise the diverse argument and clearly exposing the aim of the study. I suggest therefore the major amendment of the manuscript as developed in more details in specific comments. 

Specific comments:

-L 40-66: I suggest to restructure the whole introduction, avoiding the repetitions, bringing together the distinct facts/data, shortening the text. The idea developed in L52-54 are already exposed in lines 45-47, the idea developed in L54-57 is already present in lines 49-50, the lines 57-59 are also repetitions, lines 59-61 take up ideas already more or less exposed before, the lines 64-66 refer to H2S production already introduced line 48.

It was revised/shorten according to the reviewer suggestions.

-L64-66: There is, as far as I understand, some confusion in the whole manuscript, between the notions of “correlation”, “origins” and “consequences”. In lines 64-66 what does “included” means? Are “H2S production and biomass accumulation of intestinal SRB” only correlated to IBD or established as a cause?

The mentioned sentence was erased after revision.

-L67-68: the content of micro- and trace element is not particular 1) to SRB and 2)to cytoplasm.

The sentence was revised.

-L70-94: I suggest to restructure the whole paragraph, by 1) presenting trace elements that are cofactors (distinguishing loosely bound or prosthetic groups, metallic (Fe) or not (Mg+)); 2) presenting trace elements that are not cofactors Na+, K+ and Ca+. The latter act in the cell based on their concentration balance in the cell. All is mixed-up in the present text; 3) referring to a more extended literature. For example, why restricting the list of useful enzymes to superoxide dismutase, catalase, peroxidase ?; 4) articulating the part developed lines 83-93 in a different way since metal as cofactors, are indispensable for a lot of enzymes (as developed just before). I furthermore do not really understand the sentence as it is currently grammatically structured and I consider the use of “On the contrary” line 83, after one paragraph dedicated to metals as cofactors, as inappropriate. 5) presenting the role of trace elements in a more general way. For example, why only citing nitrogen fixation as a process influenced by metals (line 92)?

The paragraph and lines 70-94 was corrected according to the reviewer's comments. We have first described the enzyme cofactors and metals that are not cofactors playing a role in cellular transport and sulfate reduction processes. We have noted enzymes involved in the process of dissimilatory sulfate reduction and lactate consumption (lactate dehydrogenase, ATP sulfurylase, adenosine-phosphosulfate reductases). The enzymes such as peroxidase, superoxide dismutase and catalase were mentioned by us because they are enzymes responsible for stress in SRB cells.

-L95-102: Here is my main concern. I agree with the authors on the fact that “the content of trace element…may reflect the functioning of the enzyme systems.” (line 95-96) And that, consequently, “the change [in trace elements] can be caused by changes in the environment” (line 97). These changes could in fact reflect “the changes in the environments for enzymes” and thus “physiological changes in bacteria” in some IBD (lines 99-100). But the experimental approach chosen by the authors is to analyze the bacterial cell content after cultivation out of the human body, in a normalized growth medium. The trace-elements content of the bacterial cell will therefore not reflect the effect of human body conditions. The work presented could therefore not be used to establish the metabolic adaptation of SRB to IBD, but could rather be interpreted as the result of a natural selection of SRB strain in the case of IBD. The trace-element contents of the bacteria, as performed in this work, reflect the expression of the genes, in normalized conditions and not in the human body. This content therefore could be interpreted as a marker of specific genomes adapted to IBD (i.e. strain/metabolic selection), but not as a marker of a specific proteomes adapted to IBD (i.e. metabolic adaptation).

We agree with the comments of the reviewer, they are correct. However, the question of causation whether the response to changes in the cytoplasm of cells is a response to conditions that occur in patients with IBD or changes in SRB cells that cause disease is still open.

The issues of adaptation of changes in the genetic apparatus of SRB under the selective loading of the environment of the human macroorganism, explained by the esteemed reviewer, we have added to the discussion of the article. But we probably cannot yet say whether the change in the concentration of SRB cytoplasmic elements is a consequence of genetic changes (i.e. strain / metabolic selection) or a result of proteomic changes and metabolic adaptation (i.e. metabolic adaptation). This requires studies of the genome and functional genes of SRB involved in the emergence of IBD, such studies, we plan in the future, because the question is the problem of human-microbiome interaction and what is primary and what is still a secondary question.

-L209-213: First of all I do not understand the sentence, probably due to the absence of one bracket. But 1) why such variations in bacteria content from similarly infected patients?; 2) why two standard deviations (for minima and maxima). What does it mean? The Usual presentation of such data is an average mean together with a standard deviation. The “increase” or “decrease” “factors” seem to be calculated based on average mean, I therefore suggest to redraw the Fig 2 showing these average mean; 3) I would suggest to present the analysis of data in a more systematic way: for Crohn's disease patients, present the unaffected, the increased and the decreased values, and the same for UC disease patients; 4) I am not convinced by the necessity of the Table 2 in addition of Figure 2; 5) I don't understand why the author consider the Mo content not significantly affected in UC patients. I furthermore suspect an error in the value present in Table 2 for Mo.

We did some revision, but we think that the Table 2 is necessary to be present in the manuscript since it is showing in a more clear way the found statistical differences.

-L295-390: I do not really understand the link of these discussion parts with the presented data.

The obtained data represent the novel results, these results cannot be found in the present way. We tried to use the present literature findings and compare it with our results. It was hard due to not present similar experiments.

-L391-404: As already exposed I am afraid the authors over-interpreted the results. The presented data only suggest a correlation between “trace element accumulation of bacterial strain” and “microbiota of IBD patients”. This correlation could suggest, and thus be interpreted as, a specific metabolism of bacterial strain present in microbiota of IBD towards trace elements. Rather than reflecting a metabolic adaptation of the isolated strain, this could be interpreted – if the correlation is real- as a metabolic selection, since the analyses of metal content are performed after growth of isolated bacteria in normalized medium rather than directly isolated from feces.

It was revised.

-L405-408: Please could the authors develop more explicitly the interpretation of observed decrease in ferric iron and formation of H2S and insoluble sulfides?

We split the sentence into two independent sentences, in this way probably it is more understandable. Iron ions can interact with H2S and form FeS (the form of sulfide), when there is a small concentration of iron ions, there is a higher chance for the occurrence of free H2S and consequently results in a toxic environment for intestines.

Minor comments:

-L 43-45: please correct the sentence for grammar. I suggest "A significant role in the intestinal tract is played..." or "the intestinal microbiota that ...the macroorganism plays a significant role..."

It was revised.

-L46: please correct the sentence for grammar. I suggest "represent one of the important" or “an important…”

It was revised.

-L52-54: I do not understand the contradiction of the idea exposed in this sentence, implied by “though”, with the preceding one.

It was revised.

-L54: please correct the sentence for grammar. I suggest “SRB are” and since SRB is already in plural, please use “SRB” rather than “SRBs”

It was revised.

-L55, 60, 61: delete unwanted spaces

It was deleted.

-L54-57: please correct the sentence for grammar, a verb is missing.

It was revised.

-L70: I do not understand the use of “Certainly” since the authors developed in the following clear data.

It was revised.

-L75-76: Re-write “Cu-SOD” for homogeneity, and please introduce the SOD abbreviation.

It was explained.

-L74 and L80: heme contents iron, thus why distinguishing “iron ion” and “heme” in line 74? if you speak about metal, please, use iron instead of heme in line 80.

It was revised.

-L81-82: what do you mean by “transition elements”?

It was corrected.

-L114-122: The distinction between “Bacterial cultures” and “Bacterial cultivation” does not make sense to me.

It was revised.

-L128: “10 mL L-1” is not a concentration unit.

It was corrected.

-L324: I do not understand what does it mean, should we read “0.5 mM”?

It was corrected.

The manuscript has been corrected according to your comments and we kindly ask to you to accept our manuscript for the publication.

Best Regards,

Authors

Round 2

Reviewer 2 Report

The manuscript applsci-1069195_revision done-2, entitled “Basic Bioelements Content in Anaerobic Intestinal Sulfate-Reducing Bacteria” by Kushkevych et al. has been significantly improved following most of the suggestions made.

There is, however, still a number of typos, among them : P1-L2 : “occur” rather “occur””; P2-L4: “sufferinf” rather than “suffering”…, P10: “compated” rather than “compared”; P11: “to support growth its growth.”; P11: “compunds” rather than “compounds”; P12: “from the both” rather than “from both”. These typing errors leave a doubt as to the care taken by the authors.

All the part on” metals in the cell” has been improved(P2) but would still need improvement. And P6-P7, there is no significant change in the way of values presentation in Fig 2 and in the text. I still do not see what do the two variances for each trace element content mean, what the orange square means. For example, for the Ca content, I do not visualize, on fig 2 the X value given on Table 2.

But the format of the sent version of the ms (without any line and already with “citation”) let me think the editors do not wait for any future revision from me. I thus accept this version.